# Genomic diversity of 39 samples of *Pyropia* species grown in Japan

Yukio Nagano[1☺]*, Kei Kimura[2☺], Genta Kobayashi[2], Yoshio Kawamura[2]

**1** Analytical Research Center for Experimental Sciences, Saga University, Saga, Japan, **2** Faculty of Agriculture, Saga University, Saga, Japan

☺ These authors contributed equally to this work.
* nagano@cc.saga-u.ac.jp

**Data Availability Statement:** Sequences are available at the DNA Data Bank of Japan Sequence Read Archive (http://trace.ddbj.nig.ac.jp/DRASearch/submission?acc=DRA010178).

## Abstract

Some *Pyropia* species, such as nori (*P. yezoensis*), are important marine crops. We conducted a phylogenetic analysis of 39 samples of *Pyropia* species grown in Japan using organellar genome sequences. A comparison of the chloroplast DNA sequences with those from China showed a clear genetic separation between Japanese and Chinese *P. yezoensis*. Conversely, comparing the mitochondrial DNA sequences did not separate Japanese and Chinese *P. yezoensis*. Analysis of organellar genomes showed that the genetic diversity of Japanese *P. yezoensis* used in this study is lower than that of Chinese wild *P. yezoensis*. To analyze the genetic relationships between samples of Japanese *Pyropia*, we used whole-genome resequencing to analyze their nuclear genomes. In the offspring resulting from cross-breeding between *P. yezoensis* and *P. tenera*, nearly 90% of the genotypes analyzed by mapping were explained by the presence of different chromosomes originating from two different parental species. Although the genetic diversity of Japanese *P. yezoensis* is low, analysis of nuclear genomes genetically separated each sample. Samples isolated from the sea were often genetically similar to those being farmed. Study of genetic heterogeneity of samples within a single aquaculture strain of *P. yezoensis* showed that samples were divided into two groups and the samples with frequent abnormal budding formed a single, genetically similar group. The results of this study will be useful for breeding and the conservation of *Pyropia* species.

## Introduction

The genus *Pyropia* is a marine red alga belonging to the family Bangiaceae. Within the genus *Pyropia*, *P. yezoensis* Ueda (Susabi-nori in Japanese) and *P. haitanensis* Chang et Zheng (tanzicai in Chinese) are economically important marine crops that are consumed in many countries [1]. These are farmed in Japan, China and Korea. In recent years, they have been eaten as ingredients of sushi and snacks around the world, and their consumption is increasing. Its production in 2018 was 2 million tonnes [2]. Other species are cultivated or wild, and some are occasionally used as edibles. For example, *P. tenera* Kjellman (Asakusa-nori in Japanese) was once cultivated in Japan but has now been replaced by *P. yezoensis* in many places. *P. yezoensis*

**Funding:** This study was supported by the "Projects for sophistication of production and utilisation technology supporting local agriculture and marine industry" from Saga University (http://www.saga-u.ac.jp/) (to YN, KK, GK, YK) and KAKENHI (18K19235) from the Japan Society for the Promotion of Science (https://www.jsps.go.jp/) (to KK). The funders had no role in study design, data collection and analysis, decision to publish, or preparation of the manuscript.

**Competing interests:** The authors have declared that no competing interests exist.

has also been shown to have pharmacological and nutritional properties [reviewed in 3, 4]. For example, a peptide derived from this seaweed induces apoptosis in cancer cells [5,6]. The eicosapentaenoic acid-rich lipid has also been shown to be beneficial to health by alleviating hepatic steatosis [7].

Inter- and intraspecific genetic diversity of *Pyropia* has not been well studied genomically. Studying the genomic diversity of *Pyropia* species is necessary for planning the breeding and conservation of these seaweeds. This information is also important for the use of species that have not been utilized previously. Especially, utilization of underutilized species and breeding are important to address climate change. Prior to the genomic era, various studies analyzed the genetic diversity of *Pyropia* species. For example, studies used the methods based on simple sequence repeat [8,9], DNA sequencing of several genes [10–28], amplified fragment length polymorphism [29,30], and polymerase chain reaction-restriction fragment length polymorphism [31]. These previous studies analyzed small regions of the genome. In addition, many of these previous studies analyzed the genetic differences between species by phylogenetic analysis. In other words, few have analyzed population structure or admixture within species.

However, few genomic approaches were applied to study the genetic diversity of *Pyropia* species. For example, to study the genetic diversity of *P. yezoensis* in China, high-throughput DNA sequencing analyzed variations in the organellar genome [32]. This study divided wild *P. yezoensis* from Shandong Province, China, into three clusters. Several methods, such as restriction-site associated DNA sequencing [33], genotyping by sequencing [34], and whole-genome resequencing [35,36] use high-throughput DNA sequencing and can be applied to the analysis of the nuclear genome. Among them, whole-genome resequencing is the most effective way to get a comprehensive view of the entire nuclear genome. In general, whole-genome resequencing requires short reads generated by high-throughput DNA sequencing. The short reads obtained from whole-genome resequencing can be appropriate in another application, the *de novo* assembly of the organellar genome [37]. Hence, short reads are useful for research on both the nuclear and organellar genomes. Besides, whole-genome sequencing of the nuclear genome is suitable for analyzing the population structure and admixture within a species.

The Ariake sound is located in the south-west of Japan (Fig 1). It has an area of 1,700 $km^2$. It is characterized by a tidal range of up to 6 m. This tidal range is effectively used for the cultivation of seaweed. As a result, the Ariake sound is the most productive place for seaweed in Japan.

In this study, we conducted a phylogenetic analysis of 39 samples of *Pyropia* species grown in Japan using chloroplast DNA sequences assembled from short reads. To analyze the population structure and admixture, we also used whole-genome resequencing to analyze the nuclear genomes of 34 samples of *P. yezoensis*, one sample of the closely related *P. tenera*, and one sample of the offspring resulting from cross-breeding between *P. yezoensis* and *P. tenera*. Particularly, we focused on the analysis of some kinds of seaweed in the Ariake sound, Japan, where seaweed production is active.

## Materials and methods

### Materials

The collection sites were represented on a map generated with SimpleMappr [38] (Fig 1) and shown in Table 1. When the collection was performed, a single blade (a mixture of four types of haploid cell) was isolated. Next, carpospores (diploid spores) were collected from this blade and cultured. The culture of carpospores derived from single blade was defined as 'culture' in this study. Sometimes, one of the cultures was selected for aquaculture, which was defined as 'strain' in this study. A culture was also created from a strain, which is indicated in Table 1.

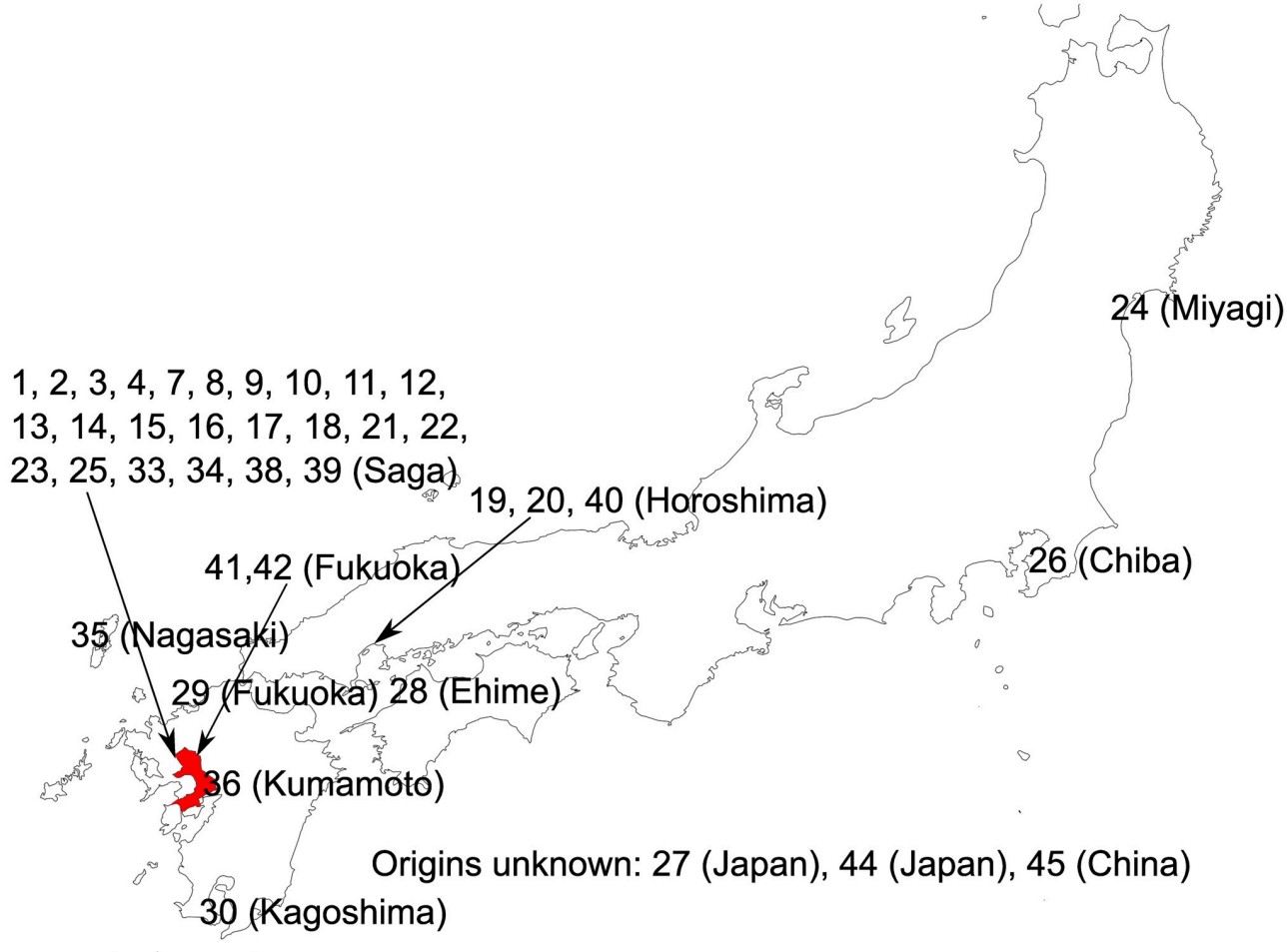

**Fig 1. Positions of the isolation sites.** The base map was produced using SimpleMappr (https://www.simplemappr.net/). The Ariake sound is shown in red.

Each culture was used as a sample in this study. In Japan, various cultures/strains are preserved in various institutions. Therefore, the acquisition of culture was not necessarily performed by the authors, so some cultures were provided by other researchers. From these cultures, this study focused on those grown in the Ariake sound.

Culture containing homozygous diploid cells derived from a single haploid cell was prepared. For the creation of this type of culture, blade was first cultured in the water of Ariake sound supplemented with modified nutrients SWM-III [39] at 17–22 ˚C under 100 μmol/m$^2$/s (11–13:13–11 h LD). Then, a monospore (a single haploid cell) was obtained from each blade according to a previous method [26]. Finally, conchocelis (homozygous diploid cells) derived from a single monospore were cultured in the water of Ariake sound supplemented with modified nutrients SWM-III at 18 ˚C under 30 μmol/m$^2$/s (11:13 h LD).

## DNA purification and sequencing

DNA was extracted from the conchocelis of each sample using the DNAs-ici!-F (Rizo, Tsukuba, Japan) according to the instructions of the manufacturer, followed by RNase A (NIPPON GENE, Tokyo, Japan) treatment. The quality of the isolated genomic DNA was checked

**Table 1. List of *Pyropia* samples.**

| Sample name | Purification to became homozygous | Species | Collection site | Isolation year | Notes |
|---|---|---|---|---|---|
| **Pyr_1** | + | *P. yezoensis* | Ariake sound, Saga Prefecture, Japan | 2010 | Culture derived from 'Shin Saga 4 gou' strain. Normal phenotype. Strain recommended by Saga Prefecture Fishery Cooperative Federation. |
| **Pyr_2** | + | *P. yezoensis* | Ariake sound, Saga Prefecture, Japan | 2010 | Culture derived from 'Shin Saga 4 gou' strain. Normal phenotype. Strain recommended by Saga Prefecture Fishery Cooperative Federation. Preservation place is different from that of Pyr_1. |
| **Pyr_3** | + | *P. yezoensis* | Ariake sound, Saga Prefecture, Japan | 2010 | Culture derived from 'Shin Saga 4 gou' strain, normal phenotype. Reisolated from Pyr_2 in 2017. |
| **Pyr_4** | + | *P. yezoensis* | Ariake sound, Saga Prefecture, Japan | 2010 | Culture derived from 'Shin Saga 4 gou' strain, normal phenotype. Reisolated from Pyr_2 in 2017. |
| **Pyr_7** | + | *P. yezoensis* | Ariake sound, Saga Prefecture, Japan | 2010 | Culture derived from 'Shin Saga 4 gou' strain, normal phenotype. Reisolated from Pyr_2 in 2017. |
| **Pyr_8** | + | *P. yezoensis* | Ariake sound, Saga Prefecture, Japan | 2010 | Culture derived from 'Shin Saga 4 gou' strain, normal phenotype. Reisolated from Pyr_2 in 2017. |
| **Pyr_9** | | *P. yezoensis* | Ariake sound, Saga Prefecture, Japan | 2010 | Culture derived from 'Shin Saga 4 gou' strain, normal phenotype. Reisolated from Pyr_2 in 2017. |
| **Pyr_10** | + | *P. yezoensis* | Ariake sound, Saga Prefecture, Japan | 2010 | Culture derived from 'Shin Saga 4 gou' strain, abnormal phenotype. Reisolated from Pyr_2 in 2017. |
| **Pyr_11** | + | *P. yezoensis* | Ariake sound, Saga Prefecture, Japan | 2010 | Culture derived from 'Shin Saga 4 gou' strain, abnormal phenotype. Reisolated from Pyr_2 in 2017. |
| **Pyr_12** | + | *P. yezoensis* | Ariake sound, Saga Prefecture, Japan | 2010 | Culture derived from 'Shin Saga 4 gou' strain, abnormal phenotype. Reisolated from Pyr_2 in 2017. |
| **Pyr_13** | + | *P. yezoensis* | Ariake sound, Saga Prefecture, Japan | 2010 | Culture derived from 'Shin Saga 4 gou' strain, abnormal phenotype. Reisolated from Pyr_2 in 2017. |
| **Pyr_14** | + | *P. yezoensis* | Ariake sound, Saga Prefecture, Japan | 2010 | Culture derived from 'Shin Saga 4 gou' strain, abnormal phenotype. Reisolated from Pyr_2 in 2017. |
| **Pyr_15** | + | *P. yezoensis* | Ariake sound, Saga Prefecture, Japan | 2010 | Culture derived from 'Shin Saga 4 gou' strain, abnormal phenotype. Reisolated from Pyr_2 in 2017. |
| **Pyr_16** | + | *P. yezoensis* | Ariake sound, Saga Prefecture, Japan | 2015 | Isolated for research purposes. Isolated from the aquaculture farm of the Saga Prefectural Ariake Fisheries Research and Development Center. |
| **Pyr_17** | + | *P. yezoensis* | Ariake sound, Saga Prefecture, Japan | 1999 | Isolated for research purposes. Isolated as probable low-temperature resistant strain. |
| **Pyr_18** | + | *P. yezoensis* | Ariake sound, Saga Prefecture, Japan | 2017 | Isolated for research purposes. Isolated from the aquaculture farm of the Saga Prefectural Ariake Fisheries Research and Development Center. |
| **Pyr_19** | + | *P. tenera* | Hiroshima Prefecture, Japan | 1978 | Isolated as *P. tenera*. Provided from Saga Prefecture Fishery Cooperative Federation. |
| **Pyr_20** | + | *P. yezoensis* | Hiroshima Prefecture, Japan | 1978 | Culture derived from marketed strain. Marketed as *P. tenera*, but morphologically similar to *P. yezoensis*. |
| **Pyr_21** | + | *P. yezoensis* | Ariake sound, Saga Prefecture, Japan | 2017 | Isolated for research purposes. Isolated from the aquaculture farm of the Saga Prefectural Ariake Fisheries Research and Development Center. |
| **Pyr_22** | + | *P. yezoensis* | Ariake sound, Saga Prefecture, Japan | 1997 | Culture derived from 'Shin Saga 1 gou' strain, strain recommended by Saga Prefecture Fishery Cooperative Federation. |
| **Pyr_23** | + | *P. yezoensis* | Ariake sound, Saga Prefecture, Japan | 2014 | Isolated for research purposes. Isolated from the aquaculture farm of the Saga Prefectural Ariake Fisheries Research and Development Center. |
| **Pyr_24** | + | *P. yezoensis* | Miyagi Prefecture, Japan | 2018 | Provided as *P. tanegashimensis* for research purposes, but morphologically similar to *P. yezoensis*. |
| **Pyr_25** | + | *P. yezoensis* | Ariake sound, Saga Prefecture, Japan | 1984 | Culture derived from 'Hagakure' strain. Strain recommended by Saga Prefecture Fishery Cooperative Federation. |
| **Pyr_26** | + | *P. yezoensis* | Chiba Prefecture, Japan | 1983 | Considered as *P. yezoensis* f. narawaensis. |

*(Continued)*

**Table 1.** (Continued)

| Sample name | Purification to became homozygous | Species | Collection site | Isolation year | Notes |
|---|---|---|---|---|---|
| Pyr_27 | + | *P. tenera × P. yezoensis* | Japan | 2005 | Culture derived from 'Gyoko strain. Crossed by National Federation of Nori & Shellfish-fishers cooperative Associations. |
| Pyr_28 | + | *P. yezoensis* | Ehime Prefecture, Japan | 1976 | Culture derived from 'Ariake 1 gou' strain. Isolated by Nagasaki University. Used in Fukuoka Prefecture. |
| Pyr_29 | + | *P. yezoensis* | Genkai Sea, Fukuoka Prefecture, Japan | 1975 | Culture derived from 'Saga 1 gou' strain. Strain recommended by Saga Prefecture Fishery Cooperative Federation. |
| Pyr_30 | + | *P. yezoensis* | Kagoshima Prefecture, Japan | 1978 | Culture derived from 'Saga 6 gou' strain. Strain recommended by Saga Prefecture Fishery Cooperative Federation. |
| Pyr_33 | + | *P. yezoensis* | Ariake sound, Saga Prefecture, Japan | 1982 | Culture derived from 'Saga 10 gou' strain. Strain recommended by Saga Prefecture Fishery Cooperative Federation named. |
| Pyr_34 | + | *P. yezoensis* | Ariake sound, Saga Prefecture, Japan | 2017 | Isolated for research purposes. Isolated from the aquaculture farm of the Saga Prefectural Ariake Fisheries Research and Development Center. |
| Pyr_35 | | *P. dentata* | Tsushima Island, Nagasaki Prefecture, Japan | 2009 | Isolated by the Saga Prefectural Ariake Fisheries Research and Development Center. |
| Pyr_36 | | *P. yezoensis* | Ariake sound, Kumamoto Prefecture, Japan | 2004 | Isolated for research purposes. Isolated from Amakusa district in Kumamoto Prefecture by Saga Prefectural Ariake Fisheries Research and Development Center. |
| Pyr_38 | + | *P. yezoensis* | Ariake sound, Saga Prefecture, Japan | 2000 | Isolated for research purposes. Green mutant. Isolated from the aquaculture farm of the Saga Prefectural Ariake Fisheries Research and Development Center. |
| Pyr_39 | + | *P. yezoensis* | Ariake sound, Saga Prefecture, Japan | 2009 | Culture derived from 'Shin Saga 3 gou' strain. Strain recommended by Saga Prefecture fishery cooperative federation. |
| Pyr_40 | | *P. yezoensis* | Hiroshima Prefecture, Japan | 1978 | Culture derived from marketed strain. Marketed as *P. tenera*, but morphologically similar to *P. yezoensis.* |
| Pyr_41 | + | *P. yezoensis* | Ariake sound, Fukuoka Prefecture, Japan | 2002 | Culture derived from 'Fukuoka Ariake 1 gou' strain. |
| Pyr_42 | + | *P. yezoensis* | Ariake sound, Fukuoka Prefecture, Japan | 1981 | Culture derived from 'Fukuoka 1 gou' strain. |
| Pyr_44 | + | *P. tenuipedalis* | Japan | 2000 | Provided by Daiichi Seimo Co., Ltd. |
| Pyr_45 | + | *P. haitanensis* | China | 2000 | Provided from Daiichi Seimo Co., Ltd. |

by 1% agarose gel electrophoresis. The concentration of DNA was determined by Qubit dsDNA BR Assay Kit (Thermo Fisher, Foster City, CA, USA).

Sequencing libraries of total DNA were generated using the NEBNext Ultra DNA library prep kit for Illumina (NEB, USA) by Novogene (Beijing, China). The libraries were sequenced with 150 bp paired-end reads using NovaSeq 6000 (Illumina, San Diego, CA, USA) by Novogene. Low-quality bases and adapter sequences from paired reads were trimmed using the Trimmomatic [40] (version 3.9) (ILLUMINACLIP:adapter_sequence:2:30:10 LEADING:20 TRAILING:20 SLIDINGWINDOW:5:20 MINLEN:50).

## Assembly-based analysis of chloroplast and mitochondrial genomes

Chloroplast and mitochondrial DNA sequences were assembled from filtered reads or public reads using the GetOrganelle [37] program (version 1.6.4). As the seed sequences for chloroplast genome assembly, other_pt.fasta included in this program was used. As the seed sequences for mitochondrial genome assembly, the complete genome sequence of *Pyropia yezoensis* (NCBI accession number: NC_017837) was used. For visual confirmation of the assembled result, filtered reads were aligned with the assembled genome by using the short

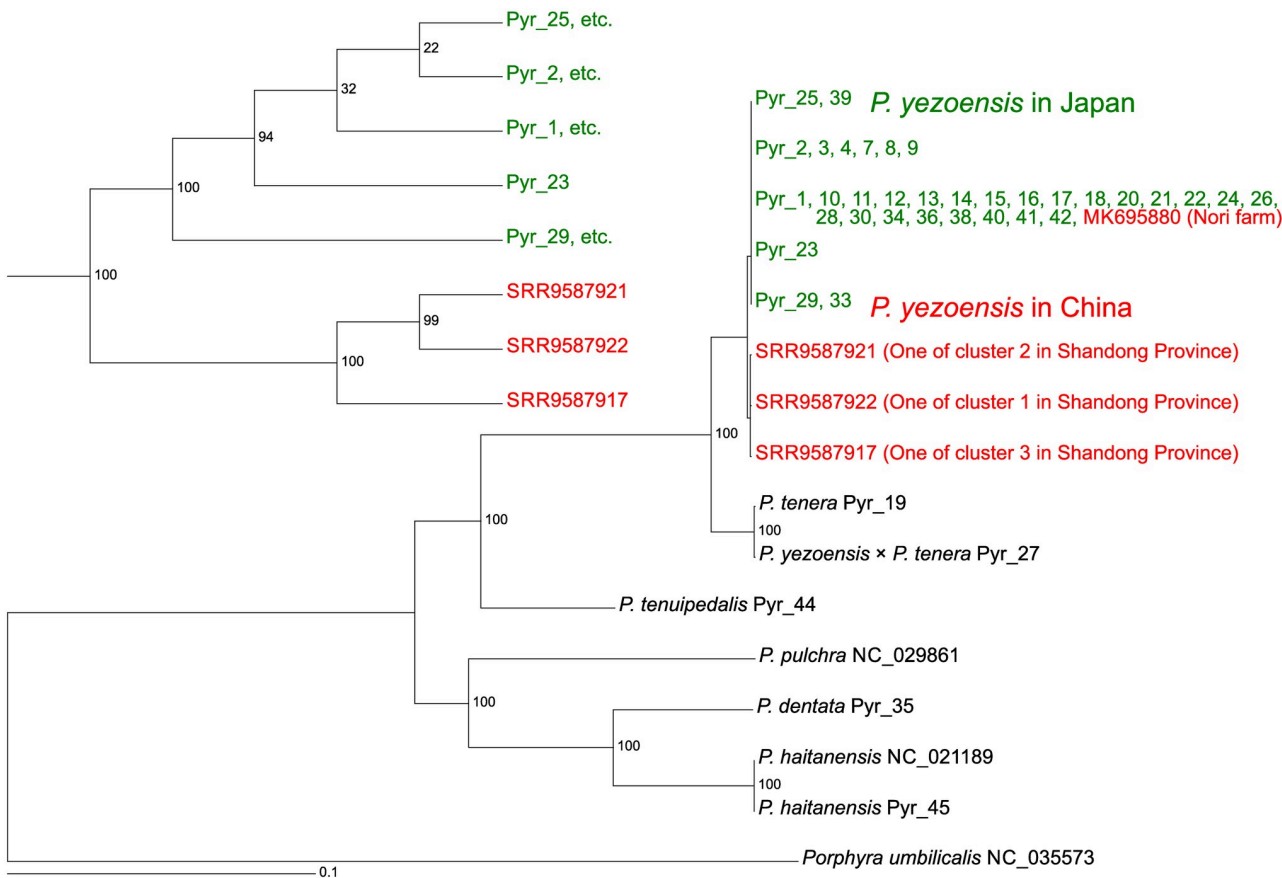

**Fig 2. Phylogenetic tree of *Pyropia* species using long single section sequences of chloroplast DNA based on maximum likelihood reconstruction.**
The numbers at the nodes indicate bootstrap values (% over 1000 replicates). The scale bar shows the number of substitutions per site. The sequence of *Porphyra umbilicalis* was used as a root. The large figure is a phylogram showing the relationships among all the data used in this analysis. The small figure in the upper left is a cladogram showing the relationships for *P. yezoensis* only. Colors were used to distinguish between Chinese and Japanese *P. yezoensis*. The parameters for RAxML were as follows: -f = a, -x = 12,345, -p = 12,345, -N (bootstrap value) = 1,000, and -m = GTRGAMMAIX).

read alignment program bowtie2 [41] (version 2.3.5.1). Samtools [42,43] (version 1.9) was used to process aligned data. The aligned data were visually inspected by the Integrative Genomics Viewer [44] (version 2.5.0). Multiple alignments of the assembled chloroplast genome were performed by MAFFT [45] (version 7.455) to create FASTA files for phylogenetic analyses. A phylogenetic tree based on the maximum likelihood method was constructed by the program RAxML [46] (version 8.2.12). ModelTest-NG was used to select the model for each analysis [47]. The parameters used in the program RAxML were shown in the legends of each figure. S1 Fig is the multi-FASTA file of the large single copy sections of chloroplast genomes used to create the phylogenetic tree in Fig 2. S2 Fig is the multi-FASTA file of the large single copy sections of chloroplast genomes used to create the phylogenetic tree in S3 Fig. S4 Fig is the multi-FASTA file of the assembled complete sequences of mitochondrial genomes used to create the phylogenetic tree in S3 Fig.

## Mapping-based analysis of nuclear, chloroplast, and mitochondrial genomes

The reference nuclear genome data of *P. yezoensis* [48] were downloaded from the National Center for Biotechnology Information (NCBI; Assembly number: GCA_009829735.1

ASM982973v1). The reference genome sequence of chloroplast was the assembled sequence of Pyr_1, one of the *P. yezoensis* samples used in this study. The reference mitochondrial genome data of *P. yezoensis* were downloaded from the NCBI (accession number: MK695879) [32]. Filtered reads of our data were aligned with the reference genome by using the program bowtie2 [41]. For the analysis of the chloroplast and mitochondrial genomes, public data of wild samples in China (NCBI Sequence Read Archive under the BioProject: PRJNA55033) were also used. Samtools [42,43] was used to process aligned data. The quality of aligned data was analyzed using the Qualimap [49] program (version 2.2.1). From aligned data, DeepVariant [50] (version 0.9.0) was used to call the variants to make vcf files [51]. Vcf files were merged using GLnexus [52] (version 1.2.6) (—config DeepVariantWGS). To process and analyze vcf files, bcftools [43] (version 1.9), bgzip implemented in tabix [43] (version 0.2.5), vcffilter implemented in vcflib [53] (version 1.0.0_rc3), and "grep" and "awk" commands of Linux were used. Low-quality data (GQ (Conditional genotype quality) < 20) were filtered out by vcffilter. The aligned data were visually inspected by the Integrative Genomics Viewer [44]. FermiKit [54] (version 0.14.dev1) was used to detect the structural variations. Multidimensional scaling (MDS) analysis was conducted based on this vcf file using the SNPRelate [55] (version 1.20.1) program. For the admixture analysis, vcf file was processed by the program PLINK [56] (version 2–1.90b3.35) (—make-bed—allow-extra-chr—recode—geno 0.1). The resulting files were used to perform the admixture analysis using the program admixture [57] (version 1.3.0).

For the phylogenetic analysis of the chloroplast, the rRNA regions were removed from the vcf file. The vcf files were converted to the FASTA format file using the VCF-kit [58] (version 0.1.6). Using FASTA file, a phylogenetic tree based on the maximum likelihood method was constructed by the program RAxML [46] (version 8.2.12). ModelTest-NG was used to select the model for each analysis [47]. The parameters used in the program RAxML were shown in the legends of each figure. S5 Fig is the multi-FASTA file of variables sites in the chloroplast genomes used to create the phylogenetic tree in S6 Fig. S7 Fig is the multi-FASTA file of variables sites in mitochondrial genomes used to create the phylogenetic tree in Fig 3.

For the phylogenetic analysis of the nuclear genome, vcf files were converted to the FASTA format file using the VCF-kit [58] (version 0.1.6) and then were converted to NEXUS format file using MEGA X [59] (version 10.0.5). Phylogenetic tree analysis was performed using SVDquartets [60] implemented in the software PAUP [61,62] (version 4.0a) using a NEXUS file. The parameters used for SVDquartets were as follows: quartet evaluation; evaluate all possible quartets, tree inference; select trees using the QFM quartet assembly, tree model; multi-species coalescent; handling of ambiguities; and distribute. The number of bootstrap analyses was 1,000 replicates. Pyr_19 was used as a root.

Public RNA sequencing data (NCBI Sequence Read Archive under accession numbers: SRR5891397, SRR5891398, SRR5891399, SRR5891400, and SRR6015124) were analyzed using HISAT2 [63] (version 2.2.0) and StringTie [64] (version 2.1.1) to determine the transcribed regions of the nuclear genome.

## Results

### Samples used in this study

Table 1 summarizes the samples used in this study, and Fig 1 shows the isolation site of each sample. We used 34 samples of *P. yezoensis*, all of which were either cultivated or isolated from close to a nori farm. Among them, 24 samples were cultures derived from the strains, which are used for aquaculture. A characteristic feature is that the 15 samples originated from a single strain 'Shin Saga 4 gou'. These 15 samples allowed for the study of genetic heterogeneity within a single strain. With seven exceptions (Pyr_20, Pyr_24, Pyr_26, Pyr_28, Pyr_29, Pyr_30, and

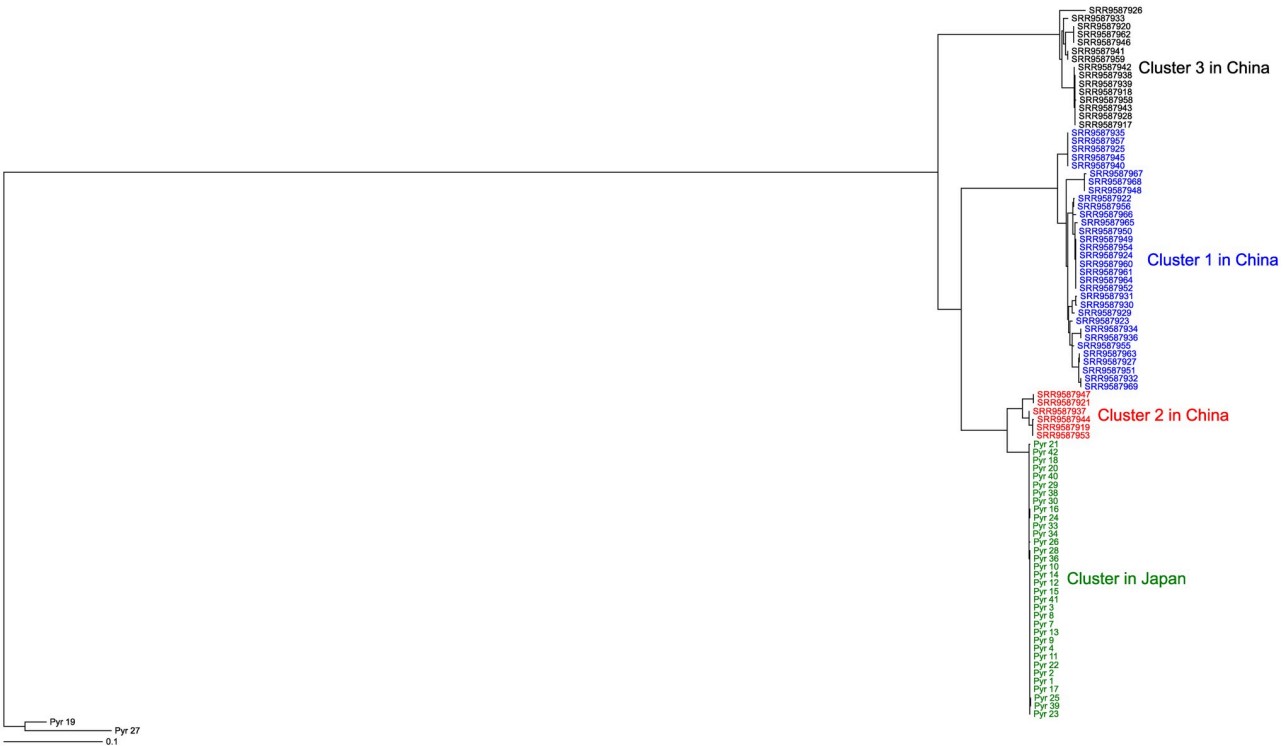

**Fig 3. Phylogenetic tree of *Pyropia yezoensis* samples using the mitochondrial DNA sequences based on maximum likelihood reconstruction.** The scale bar shows the number of substitutions per site. The sequences of Pyr_19 (*P. tenera*) and the hybrid (Pyr_27) between *P. yezoensis* and *P. tenera* were used as roots. Colors were used to distinguish between Japanese *P. yezoensis* and 3 clusters of Chinese *P. yezoensis*. The parameters for RAxML were as follows: -f = a, -x = 12,345, -p = 12,345, -N (bootstrap value) = 1,000, -c = 1, and -m = GTRCATX). Only variable sites were used in the analysis.

Pyr_40), 27 of them originated from the Ariake sound. In addition to 34 samples of *P. yezoensis*, we used *P. haitanensis* (Pyr_45), *P. dentata* Kjellman (Pyr_35), *P. tenuipedalis* Miura (Pyr_44), *P. tenera* (Pyr_19), and the hybrid between *P. yezoensis* and *P. tenera* (Pyr_27) (the offspring resulting from cross-breeding between *P. yezoensis* and *P. tenera*). In this study, we prepared 35 cultures, each of which contains homozygous diploid cells derived from a single haploid cell.

## Sequencing

The libraries from these samples were sequenced with 150 bp paired-end reads. The Number of reads ranged from 64 million to 110 million (S1 Table). These reads were used for two purposes: 1) *de novo* assembly of the chloroplast and mitochondrial genomes, and 2) mapping to a reference genome.

## Chloroplast and mitochondrial genomes

To elucidate the relationships among the 39 samples, we attempted to determine the whole chloroplast DNA sequences. Of the 39 total cases, we successfully assembled the whole chloroplast genome of 30 samples, and 9 cases (Pyr_16, Pyr_18, Pyr_21, Pyr_29, Pyr_23, Pyr_25, Pyr_28, Pyr_33, and Pyr_34) were unsuccessful. The chloroplast genomes of *Pyropia* species contain two direct repeats carrying ribosomal RNA (rRNA) operon copies [32,65,66]. The circular chloroplast genome has a large single-copy section and a short single-copy section between two rRNA repeats. The presence of two non-identical direct repeats carrying two

different rRNA operons are general features of *Pyropia* chloroplast genomes [32,65,66]. Reflecting the presence of two non-identical direct repeats, mapping of the short reads to the assembled chloroplast genome detected two types of reads in the rRNA operons (S8 Fig) in all 39 cases. The presence of two types of reads in the rRNA operons might cause the unsuccessful assembly of the whole chloroplast genome of the 9 samples. Indeed, we succeeded in assembling the long single section sequences of all 39 samples. We detected the presence of two chloroplast genomes, termed heteroplasmy, only in the sample Pyr_45 (*P. haitanensis*) (S9 Fig).

Previous research reported that wild samples of *P. yezoensis* from Shandong Province, China, can be classified into three clusters [32]. We also assembled the chloroplast genome of one sample of each cluster. By using the assembled sequences of large single copy sections and publicly available chloroplast sequences, we created a phylogenetic tree based on the maximum likelihood method (Fig 2). The number of parsimony informative sites was 24,914. The chloroplast sequences of all Japanese samples of *P. yezoensis* were very similar to each other. The Chinese wild samples of *P. yezoensis* were genetically similar to but clearly separated from the Japanese samples. The chloroplast sequence of a Chinese cultivar (MK695880) collected from the nori farm in Fujian province was identical to that of some Japanese samples but not identical to those of the Chinese wild samples. This suggests that this Chinese cultivar was introduced from Japan, where the cultivation of *P. yezoensis* was established.

We mapped short reads of *P. yezoensis* samples from Japan and China to the chloroplast genome and constructed the maximum likelihood tree using the mapped data (S6 Fig). As similar to Fig 2, Japanese and Chinese chloroplast genomes were clearly separated. In addition, chloroplast sequences of Chinese *P. yezoensis* did not clearly separate them into three clusters. Chloroplast sequences showed that the genetic diversity of Japanese *P. yezoensis* used in this study is slightly lower than that of the Chinese wild *P. yezoensis*. We applied a similar approach to the mitochondrial genome (Fig 3). Mitochondrial sequences separated Chinese *P. yezoensis* into three clusters as reported previously [32]. Thus, mitochondrial sequences, rather than chloroplast sequences, are responsible for clustering in the previous report [32]. The analysis did not separate Japanese from Chinese *P. yezoensis*. Rather, Japanese *P. yezoensis* is similar to cluster 2 of Chinese *P. yezoensis*. Importantly, mitochondrial sequences showed that the genetic diversity of Japanese *P. yezoensis* used in this study is significantly lower than that of Chinese wild *P. yezoensis*.

There were differences in topological structures between the chloroplast phylogenetic trees (Fig 2 and S6 Fig) and the mitochondrial phylogenetic tree (Fig 3); in the chloroplast phylogenetic trees, the Japanese and Chinese *P. yezoensis* were clearly separated, but in the mitochondrial phylogenetic tree, the two were not separated. This may be due to differences in the rate of evolution between the chloroplast and mitochondrial genomes. Comparisons of the phylogenetic trees showed that the rate of evolution of the mitochondrial genome was about 10 times faster than that of the chloroplast genome (S3 Fig).

## Nuclear genomes

The chloroplast and mitochondrial sequences did not clearly discriminate each sample of Japanese *P. yezoensis*. The large size of the nuclear genome makes it easy to detect genetic differences. Therefore, comparisons based on the nuclear genome are important. We analyzed the nuclear genome of only *P. yezoensis* from Japan because the Chinese *P. yezoensis* have small amounts of published reads. We mapped the short reads from 39 samples to the reference genome of *P. yezoensis*. S1 Table summarizes the results. The mean coverage of Pyr_35 (*P. dentata*), Pyr_44 (*P. tenuipedalis*), and Pyr_45 (*P. haitanensis*) was 2.7 ×, 8.8 ×, and 4.2 ×, respectively. As described above, the analysis of the chloroplast genome showed that these three samples were

not similar to *P. yezoensis*, which was reflected in the low mean coverage. Therefore, we did not use the data from these three samples for further analysis based on the mapping. In contrast, the mean coverage of Pyr_19 (*P. tenera*) and Pyr_27, the hybrid between *P. yezoensis* and *P. tenera*, was 47.4 × and 25.7 ×, respectively. Therefore, we used the data of these two samples for the subsequent analysis. The mean coverages for *P. yezoensis* ranged from 14.6 × to 83.4 ×. Before DNA extraction, we did not remove the bacteria co-cultured with seaweed. Differences in the extent of the range of mean coverage reflect the amount of these bacteria.

We called genetic variants from the mapping data. After merging the variant data of 36 samples, we filtered out low-quality data (GQ (Conditional genotype quality) < 20). The data included 697,892 variant sites. Subsequently, we analyzed variant information by MDS analysis (S10 Fig). Samples of *P. yezoensis* were clustered together. Observations of the mapping results suggests that there were very few regions that have been incorrectly mapped with bacterial sequences, but we cannot exclude the possibility that bacterial contamination may have influenced the results shown in S10 Fig.

## The hybrid between *P. yezoensis* and *P. tenera*

The MDS analysis separated Pyr_27 from the Pyr_19 (*P. tenera*), although these two samples were similar in chloroplast sequences. The analysis located the hybrid at approximately the middle position between two species, which supports the record that the hybrid was created by an interspecific cross between two species.

The previous analysis using a small number of genes reported that the hybrid between *P. yezoensis* and *P. tenera* is allodiploid in the blade cell [67,68]. In other words, if the haploid genome of *P. yezoensis* and *P. tenera* is A and B, respectively, the blade cell of the hybrid has the allodiploid genome of AB, and the conchocelis cell of the hybrid has the allotetraploid genome of AABB. We reanalyzed this possibility using the whole genome data of Pyr_1 (*P. yezoensis*), Pyr_19 (*P. tenera*), and their hybrid Pyr_27. Both Pyr_1 and Pyr_19 samples should be homozygous diploid as they are derived from a single haploid cell. The sample Pyr_27 is also derived from a single haploid cell, but if it is an allotetraploid, the genotype will be observed as heterozygous in the genome viewer. Fig 4A shows a representative example of a comparison of genotypes among three samples. This figure showed that Pyr_27 has heterozygous-like data. To further analyze the tendency, we calculated the number of the types of loci in Pyr_27 when the Pyr_1 locus carries two reference alleles (homozygous locus) and when the Pyr_19 locus carries two alternate alleles (alternative homozygous locus) (Table 2). In principle, this calculation method effectively excludes the regions that have been mapped incorrectly with bacterial sequences. In about 90% of the cases, Pyr_27 had one reference allele and one alternate allele (heterozygous locus). Therefore, Pyr_27 must be an allotetraploid. Previously, nuclear rRNA genes have shown to become homozygous after conjugation [67,68]. Indeed, in this region, the genotypes of Pyr_27 were identical to those of Pyr_19 (*P. tenera*) (Fig 4B).

## Genomic diversity of Japanese *P. yezoensis*

Of 34 samples of *P. yezoensis*, 31 samples were derived from a single haploid cell, so they should be homozygous diploid. Therefore, we analyzed these 31 samples. From the variant data, we removed loci containing potentially heterozygous loci, which may be created due to the presence of non-identical repeats. The removal of such loci can remove loci that have been mapped incorrectly with bacterial sequences. This data included 69,918 variant sites. We then performed an MDS analysis (Fig 5) and an admixture analysis (Fig 6 and S11 Fig) using this variant data. For the phylogenetic analysis, we created the tree using the data from 32 samples, including 31 samples of *P. yezoensis* and 1 sample of *P. tenera* as a root, all of which should be

A

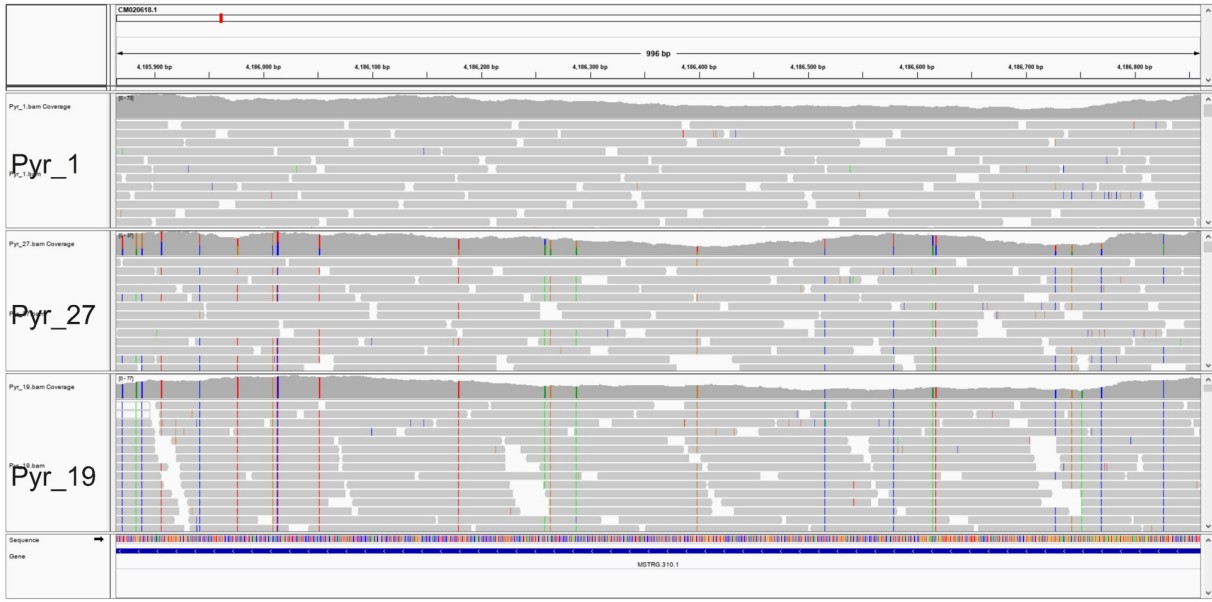

B

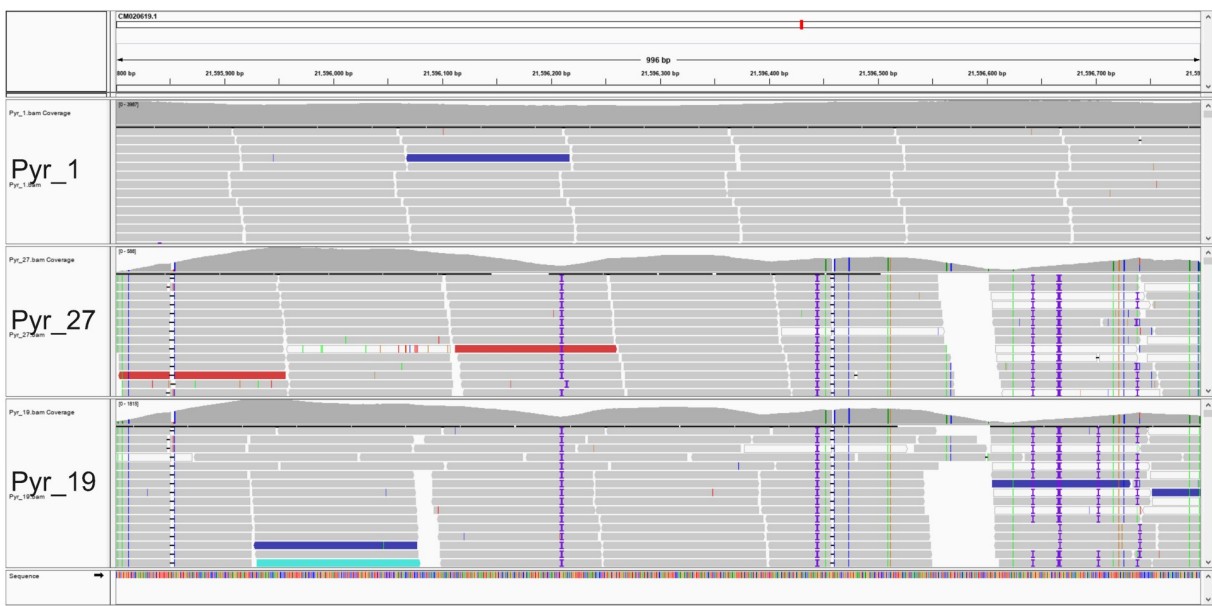

**Fig 4. Comparison of the nuclear genotypes of Pyr_1 (*Pyropia yezoensis*), Pyr_27 (the hybrid between *P. yezoensis* and *P. tener*a), and Pyr_19 (*P. tenera*) visualized by the Integrative Genomics Viewer.** (A) A representative example of allotetraploid formation. (B) Nuclear rRNA regions.

homozygous. After removal of heterozygous loci, the data included 681,115 variant sites, of which 33,985 were parsimony informative sites. We constructed a phylogenetic tree using the program SVDQuartet [60] (Fig 7). Under the coalescence model, this program assumes that intragenic recombination exists.

**Table 2. Number of loci in Pyr_27, when Pyr_1 locus carries two reference alleles and Pyr_19 locus carries two alternate alleles.**

| Two reference alleles | One reference allele/one alternate allele | Two alternate alleles |
|---|---|---|
| 18,484 | 139,259 | 141 |

Although the analyses clearly separated each sample based on genetic distance, there were three closed clusters: cluster A containing 19 samples (Pyr_1, 2, 3, 4, 7, 8, 10, 11, 12, 13, 14, 15, 17, 22, 23, 25, 28, 39, and 41), cluster B containing 2 samples (Pyr_21 and 38), and cluster C containing 2 samples (Pyr_16 and 24) (Fig 5). The phylogenetic tree detected these closed clusters with high bootstrap supports (Fig 7).

Fig 6 and S11 Fig show the results of the admixture analysis. Among the possible values of *K* (the number of ancestral populations), *K* = 2 was the most likely because *K* = 2 had the smallest cross-validation error (S11 Fig). Thus, the number of ancestral populations could be 2, with the 19 samples shown in blue in Fig 6 belonging to one population and the six samples shown in red in Fig 6 belonging to a second population. The former 19 samples were cluster A, as described above. Six samples, Pyr_18, 20, 21, 29, 33, and 38 might belong to an admixture

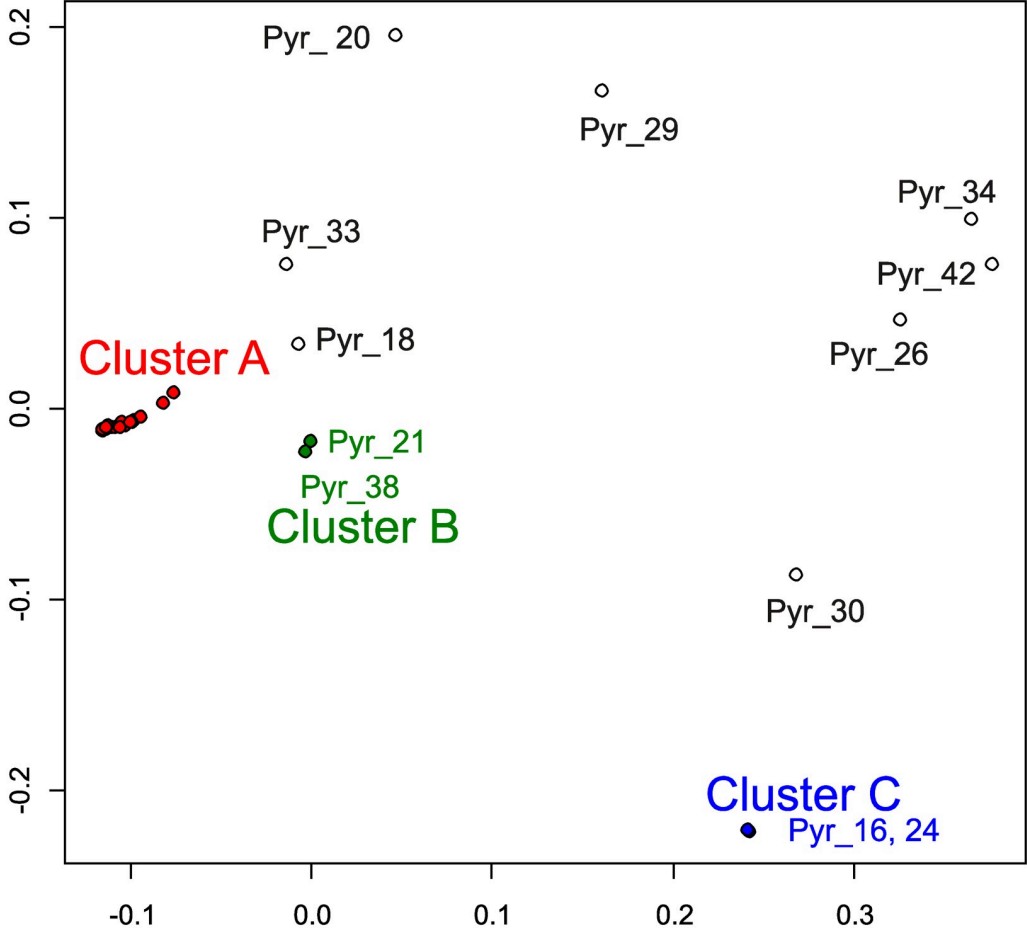

**Fig 5. Multidimensional scaling representation using nuclear DNA data of 31 *Pyropia yezoensis* samples.** Two-dimensional data were obtained in this analysis. Colors were used to show 3 clusters, cluster A, B, and C, found in this study.

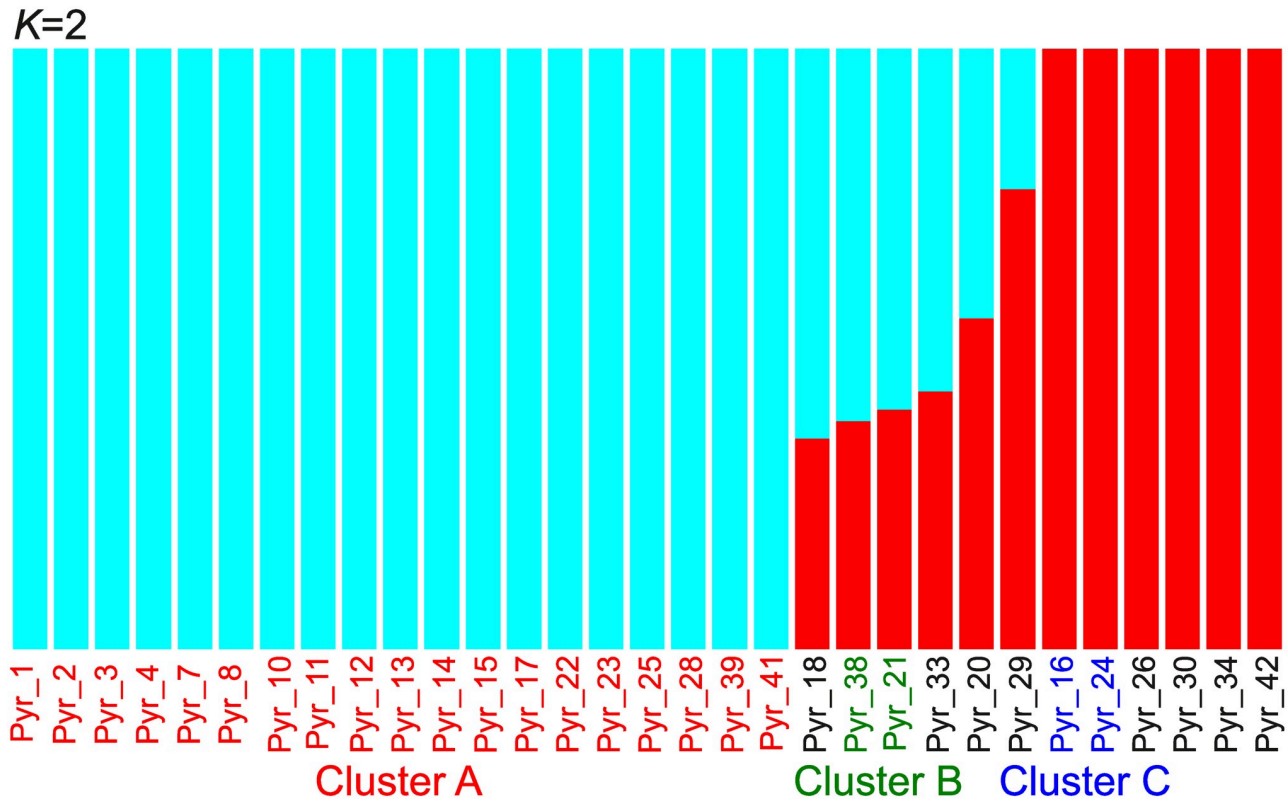

**Fig 6. Admixture analysis using nuclear DNA data of 31 *Pyropia yezoensis* samples.** The number of populations (K) was set to 2. Colors of the sample names were used to show 3 clusters, cluster A, B, and C.

population formed by the hybridization between these two ancestral populations. However, it is also possible that K is 3 or 5 (S11 Fig). Because K = 1 had the highest cross-validation error, admixture has likely happened in these 39 samples.

Because 19 samples belonging to cluster A were closely related, it was difficult to distinguish these samples in the above MDS analysis shown in Fig 5. Therefore, we analyzed only these 19 samples. The variant data from these 19 samples included 36,459 variant sites. We used this data for MDS analysis (Fig 8). Twelve samples (Pyr_1, 2, 3, 4, 7, 8, 10, 11, 12, 13, 14, and 15) are samples isolated from the single strain 'Shin Saga 4 gou', which may contain heterogeneous cells. The 'Shin Saga 4 gou' strain has two subgroups: a subgroup containing seven samples of Pyr_1, 10, 11, 12, 13, 14, and 15 and a subgroup containing five samples of Pyr_2, 3, 4, 7, and 8. Phylogenetic analysis (Fig 7) supported this observation with high bootstrap values. Among the 'Shin Saga 4 gou' strain, Pyr_10, 11, 12, 13, 14, and 15 have an abnormal phenotype. In normal samples, the rate of abnormal budding is about 15%, whereas in abnormal samples it is about 40%. These 6 samples formed the closely related cluster and were separated from Pyr_1, although these 7 samples belonged to the former subgroup. In addition to the members of strain 'Shin Saga 4 gou', Pyr_17, 22, 23, 25, 28, 39, and 41 belonged to a clade containing 19 samples. Of these, Pyr_17, 22, 23, 25, 39, and 41 were cultivated or isolated in the Ariake sound. Pyr_28 was from the sound in Ehime Prefecture.

To detect variants specific to abnormal samples (Pyr_10, 11, 12, 13, 14, and 15), we extracted 69 candidate loci by analyzing variant call data and then further selecting the candidates by visual inspection (S2 Table). We also used the program FermiKit [54] to detect structural variants specific to abnormal samples but failed to detect them. Furthermore, Pyr_36 is a

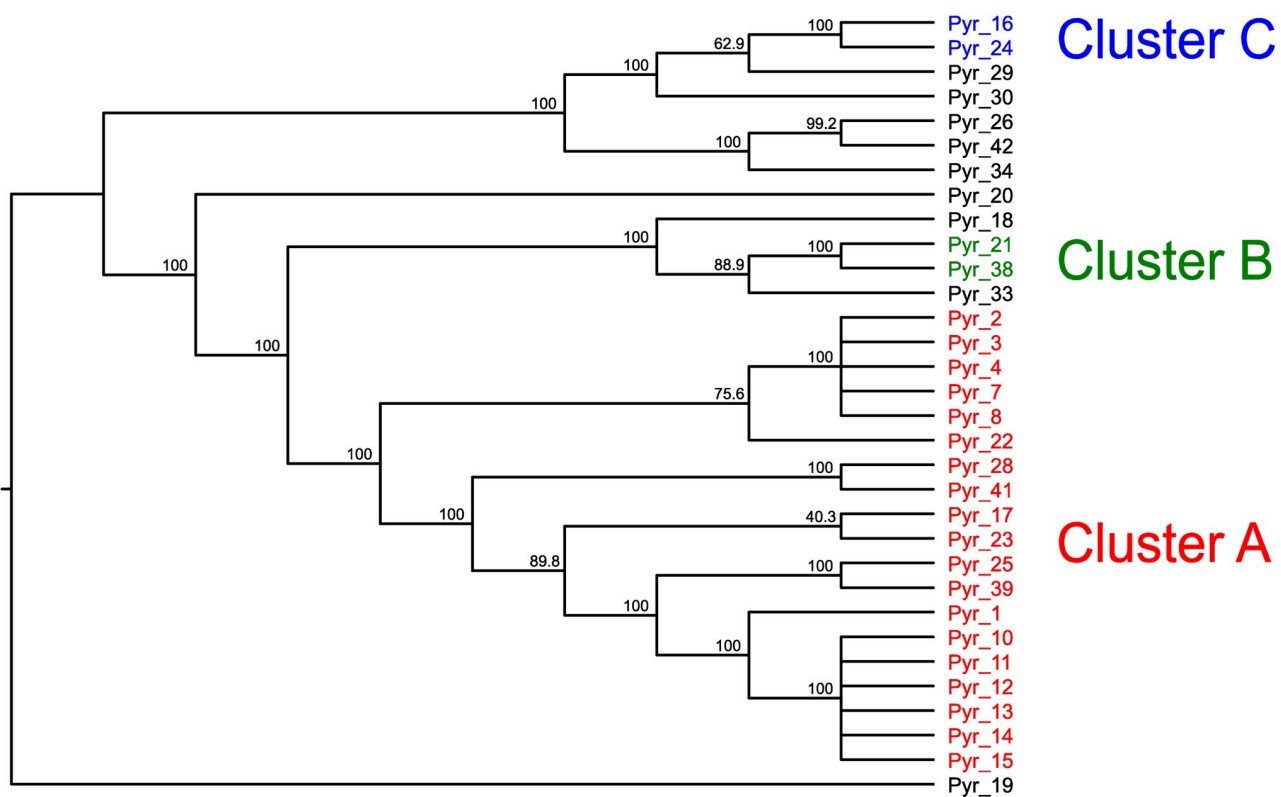

**Fig 7. Phylogenetic tree constructed using the SVDquartets with PAUP using nuclear DNA data of 31 *Pyropia yezoensis* samples and 1 *P. tenera*.** The numbers at the nodes indicate bootstrap values (% over 1,000 replicates). The data of Pyr_19 (*P. tenera*) was used as a root. Colors of the sample names were used to show 3 clusters, cluster A, B, and C.

green mutant. The similar strategies detected 104 candidate loci (S3 Table). In both cases, some of the loci were located within the gene or close to the gene. For example, deletion is detected in the calmodulin gene in the green mutant.

## Discussion

A comparison of Japanese and Chinese samples of *P. yezoensis* revealed that the diversity of chloroplast and mitochondrial sequences in Japan is lower than that in China (Figs 2 and 3 and S6 Fig). This is especially evident in the case of mitochondrial DNA, which may be because most of the Japanese *P. yezoensis* used in this analysis were either cultivated or isolated from close to a nori farm. Therefore, it is important to investigate Japanese wild samples, especially from sounds where farming is not performed, in the future. Since the farming of *P. yezoensis* has begun in Japan, studying their diversity is an attractive topic. The chloroplast and mitochondrial sequences did not clearly discriminate each sample of Japanese *P. yezoensis*, probably because the size of them is smaller than that of nuclear genome. Indeed, in this study, we were able to distinguish the Japanese samples by analyzing the nuclear genomes.

The phylogenetic tree of chloroplasts (Fig 2) is in good agreement with previous studies. Among *Pyropia* species, *P. tenera* is the closest relative to *P. yezoensis* at the DNA level [26–28]. In addition, they can be crossed with each other [69]. In fact, Pyr_19 (*P. tenera*) was present on a branch next to *P. yezoensis*. The nucleotide sequence of the hybrid (Pyr_27) between *P. yezoensis* and *P. tenera* was more similar to that of *P. tenera* than that of *P. yezoensis*. Examination of the sequence revealed that, as expected, Pyr_45 was *P. haitanensis*. The analysis

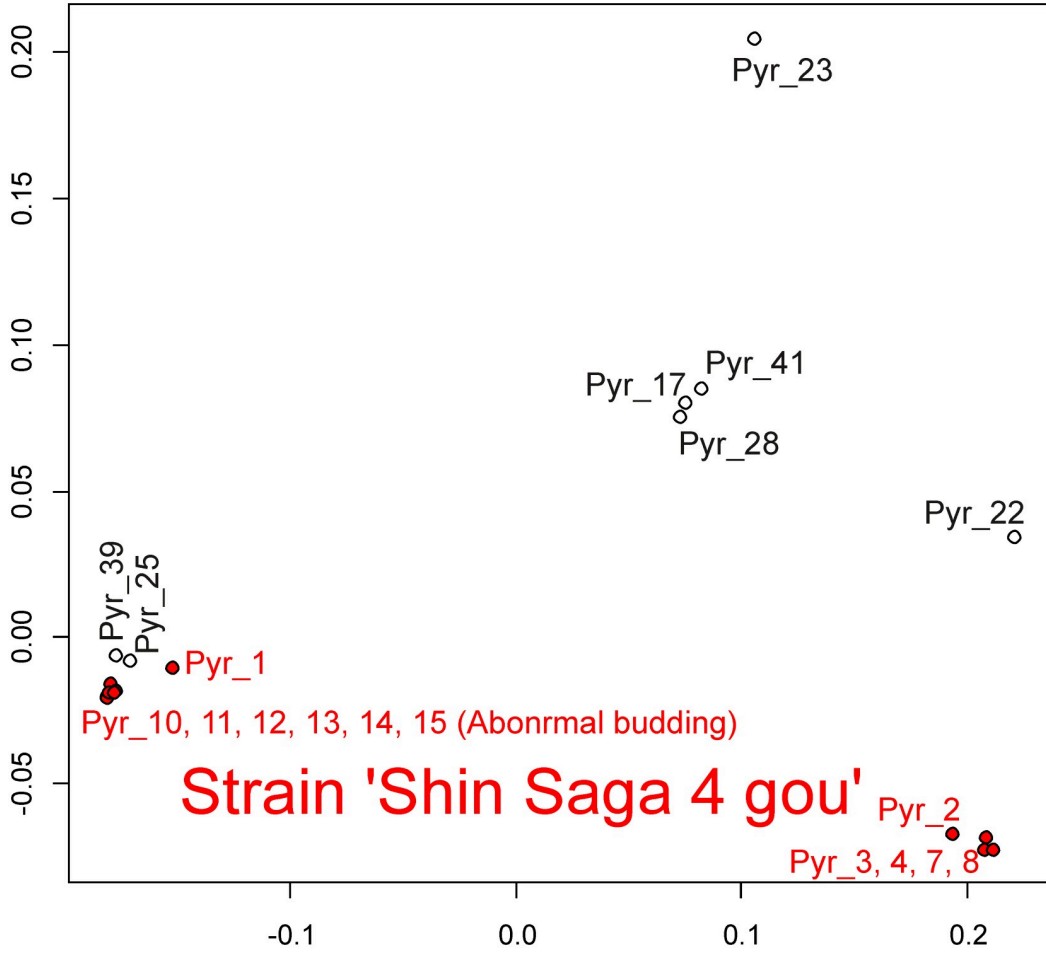

**Fig 8. Multidimensional scaling representation using nuclear DNA data of 19** *Pyropia yezoensis* **samples belonging to the cluster A.** Two-dimensional data were obtained in this analysis. Members of the strain 'Shin Saga 4 gou' were shown in red.

showed that Pyr_35 (*P. dentata*) is similar to *P. haitanensis*, as in previous studies [28]. Pyr_44 (*P. tenuipedalis*) belonged to a clade containing *P. yezoensis*, which is also consistent with previous molecular analyses [28].

We examined the genetic changes that have occurred during an interspecific cross between *P. yezoensis* and *P. tenera*. Both chloroplast and mitochondrial genomes of the hybrid were similar to those of *P. tenera* (Figs 2 and 3 and S6 Fig). At least in this one conjugation event, a mechanism of maternal, paternal, or random DNA transmission might eliminate the chloroplast and mitochondrial genomes of *P. yezoensis*, one of two parental chloroplast genomes. However, there is the possibility that chloroplasts and mitochondria may have different inheritance mechanisms.

The changes that have occurred in the nuclear genome during an interspecific cross are more interesting. MDS analysis (S10 Fig) located the hybrid at approximately the middle position between two species. This is due to the allotetraploid formation and not due to the DNA recombination between two species. We analyzed the loci in the hybrid and found that when *P. yezoensis* had homozygous locus and the *P. tenera* had alternative homozygous locus, about 90% of the loci were allotetraploid (Table 2). Thus, to our knowledge, the current study

demonstrated the allotetraploid formation at the genomic level for the first time, although the analysis of a small number of genes showed that the interspecific cross between *P. yezoensis* and *P. tenera* could produce an allotetraploid [67,68]. The creation of interspecific hybrids is an important method in the breeding of seaweeds. The methods used in this study will be useful in the analysis of interspecific hybridization after the conjugation because we can track changes in the nuclear genome. After the interspecific cross, the region of the nuclear rRNA became homozygous (Fig 4B). This was consistent with previous studies [67,68]. Thus, a mechanism to eliminate one of the two types of nuclear rRNA genes is present. In contrast, there were two types of rRNA genes in the chloroplast genome.

The hybrid is slightly more similar to *P. yezoensis* than to *P. tenera* in the MDS analysis (S10 Fig). Furthermore, we analyzed the loci in the hybrid and found that when the *P. yezoensis* had homozygous locus and the *P. tenera* had alternative homozygous locus, about 10% of the loci were identical to those of *P. yezoensis* (Table 2). In order to explain this observation, we must consider two possibilities. One possibility is that this study did not use the real parents of the hybrid. The *P. tenera* used in the cross might be a more genetically similar sample to *P. yezoensis*. Related to this possibility, the mitochondrial sequence of the hybrid was similar to but not identical to that of *P. tenera* used in this study (Fig 3). The second possibility is that recombination or other mechanisms eliminated the sequence of *P. tenera* drastically. Considering the two possibilities, it is important to analyze the genomes of the trio, the hybrid and its biological parents, immediately after performing an interspecific cross.

There were three closed cluster in MDS analysis (Fig 5) and phylogenetic analysis (Fig 7). The members of cluster B, Pyr_21 and 38, were independently isolated from an aquaculture farm of the Saga Prefectural Ariake Fisheries Research and Development Center, but they originated in the same sound. In cluster C, Pyr_16 was isolated from an aquaculture farm at the Saga Prefectural Ariake Fisheries Research and Development Center, and Pyr_24 was provided as *P. tanegashimensis* previously. However, Pyr_24 was morphologically similar to *P. yezoensis*. For this reason, it appeared that the sample had been misplaced previously. Most of the samples used in this study were cultured or isolated in the Ariake sound, which is surrounded by Fukuoka, Saga, Nagasaki, and Kumamoto Prefectures (Fig 1), whereas Pyr_20, Pyr_24, Pyr_26, and Pyr_30 were from the sound in Hiroshima, Miyagi, Chiba, and Kagoshima Prefectures, respectively. In addition, Pyr_29 was from the Genkai Sea in Fukuoka Prefecture. (The Genkai Sea in northern Fukuoka Prefecture and the Ariake sound in southern Fukuoka Prefecture are separated by land.) Thus, the analyses did not clearly separate the seaweeds of other places from those of the Ariake sound. Pyr_20 was once marketed as *P. tenera*, but morphologically it is more likely to be *P. yezoensis*. Our genetic analysis confirmed that Pyr_20 is *P. yezoensis*.

Our analysis revealed that genetically similar seaweeds were frequently used in the Ariake sound (Figs 5 and 8), although this fact had been unknown previously. Furthermore, it turns out that genetically similar seaweed had been repeatedly isolated by researchers. Cluster A is a typical example of the genomic similarity because cluster A contains samples that could not be considered an 'Shin Saga 4 gou' strain. Of the 19 ascensions of cluster A, all except one were cultivated in the Ariake sound or isolated from the Ariake sound. This may be a problem for the conservation of *P. yezoensis*. We need to examine whether the genetic diversity of *P. yezoensis* is maintained in sounds where farming is not performed.

Of the 19 ascensions of cluster A, Pyr_28 was isolated in the sound in Ehime Prefecture. It is possible that this was due to human activity. Seaweed farming began in the Ariake sound later than other areas. Therefore, ancestors of cluster A members may have been artificially brought into the Ariake sound from the other areas such as Ehime Prefecture. Seaweed farming is now very active in the Ariake sound. Therefore, we do not exclude the possibility that

Pyr_28 was artificially brought into Ehime Prefecture from the Ariake sound and isolated in Ehime Prefecture. In any case, the actual situation will not be known until we analyze the genomes of various seaweeds in various parts of Japan.

The samples in the 'Shin Saga 4 gou' strain were very similar. Despite the similarities, we were able to detect genetic differences among these. In other words, we were able to detect heterogeneity in the strain successfully. It is an interesting finding, for example, that the samples having abnormal budding are in tight clusters. Six samples with frequent abnormal budding formed a single, genetically similar closed cluster. These abnormal samples probably diverged after an event in which a single mutation occurred. The heterogeneity of somatic cells has been studied in cancer research [70]. Similar studies are now possible in seaweed.

In this study, we detected variants specific to abnormal samples (S2 Table) or to a green mutant (S3 Table). However, a number of candidate loci were present. Further studies are needed to identify the responsible genes. Targeted deletion by genome editing is one of the ways to identify them.

In summary, we used high-throughput sequencing to examine the genomic diversity of *Pyropia* species. To our knowledge, this is the first study to examine the genomic diversity of *Pyropia* species at the genome level. The information obtained in this study could be used to develop a breeding and conservation plan. A variety of *Pyropia* species grow wild or are cultivated in Japan and around the world. It is essential to perform genomic studies of these seaweeds.

## Supporting information

**S1 Fig. The multi-FASTA file of the large single copy sections of chloroplast genomes used to create the phylogenetic tree in Fig 2.**
(PDF)

**S2 Fig. The multi-FASTA file of the large single copy sections of chloroplast genomes used to create the phylogenetic tree in S3 Fig.**
(PDF)

**S3 Fig. Comparison of phylogenetic trees between the chloroplast and mitochondrial DNA sequences.** Phylogenetic trees were constructed based on maximum likelihood method. The DNA sequences from the large single copy sections of chloroplast genomes were used to create a chloroplast phylogenetic tree. The DNA sequences from the assembled sequences of mitochondrial genomes were used to create a mitochondrial phylogenetic tree. The numbers at the nodes indicate bootstrap values (% over 1000 replicates). The scale bar shows the number of substitutions per site. In each analysis, the midpoint was used as a root. The parameters for RAxML in the analysis of chloroplast DNA sequences were as follows: -f = a, -x = 12,345, -p = 12,345, -N (bootstrap value) = 1,000, -c = 1 and -m = GTRCATX). The parameters for RAxML in the analysis of mitochondrial DNA sequences were as follows: -f = a, -x = 12,345, -p = 12,345, -N (bootstrap value) = 1,000, and -m = GTRGAMMAX).
(PDF)

**S4 Fig. The multi-FASTA file of the assembled complete sequences of mitochondrial genomes used to create the phylogenetic tree in S3 Fig.**
(PDF)

**S5 Fig. The multi-FASTA file of variables sites in the chloroplast genomes used to create the phylogenetic tree in S6 Fig.**
(PDF)

**S6 Fig. Phylogenetic tree of *Pyropia yezoensis* samples using the chloroplast DNA sequences based on maximum likelihood reconstruction.** The scale bar shows the number of substitutions per site. The sequences of Pyr_19 (*P. tenera*) and the hybrid (Pyr_27) between *P. yezoensis* and *P. tener*a were used as roots. Colors were used to distinguish between Japanese *P. yezoensis* and 3 clusters of Chinese *P. yezoensis*. The parameters for RAxML were as follows: -f = a, -x = 12,345, -p = 12,345, -N (bootstrap value) = 1,000, -c = 1, and -m = GTRCAT). Only variable sites were used in the analysis.
(PDF)

**S7 Fig. The multi-FASTA file of variables sites in mitochondrial genomes used to create the phylogenetic tree in Fig 3.**
(PDF)

**S8 Fig. Genotypes of the rRNA repeats of the chloroplast genome of Pyr_1 (*Pyropia yezoensis*) visualized using the Integrative Genomics Viewer.**
(PDF)

**S9 Fig. Genotypes of the chloroplast genome of Pyr_45 (*Pyropia haitanensis*) visualized using the Integrative Genomics Viewer.**
(PDF)

**S10 Fig. Multidimensional scaling representation using nuclear DNA data of 34 *Pyropia yezoensis* samples, Pyr_19 (*P. tenera*), and Pyr_27 (the hybrid between *P. yezoensis* and *P. tener*a).** Two-dimensional data were obtained in this analysis. Colors were used to show *P. yezoensis* samples, *P. tenera*, and the hybrid between *P. yezoensis* and *P. tener*a.
(PDF)

**S11 Fig. The graph of *K* values vs cross-validation errors (upper) and admixture analysis using nuclear DNA data of 31 *Pyropia yezoensis* samples (*K* = 3, 4, 5).** Colors of the sample names were used to show 3 clusters, cluster A, B, and C.
(PDF)

**S1 Table. Summary of the quality of aligned data analyzed by the Qualimap program.**
(XLSX)

**S2 Table. List of variants specific to abnormal samples.**
(XLSX)

**S3 Table. List of variants specific to green mutant.**
(XLSX)

## Acknowledgments

We would like to thank H. Matsuo for technical assistance with the experiments. We would also like to thank Daiichi Seimo Co. Ltd., Saga Prefectural Ariake Fisheries Research and Development Center, Ariake Regional Laboratory of Fukuoka Fisheries and Marine Technology Research Center, Saga Prefectural Ariake Fishery Cooperative, and the National Federation of Nori & Shellfish-fishers cooperative associations for providing us with the samples of *Pyropia* species used in this study. We would like to thank Editage (www.editage.com) for English language editing.

## Author Contributions

**Conceptualization:** Yukio Nagano, Kei Kimura, Yoshio Kawamura.

**Data curation:** Yukio Nagano, Kei Kimura, Yoshio Kawamura.

**Formal analysis:** Yukio Nagano, Kei Kimura, Yoshio Kawamura.

**Funding acquisition:** Yukio Nagano, Kei Kimura, Genta Kobayashi, Yoshio Kawamura.

**Investigation:** Yukio Nagano, Kei Kimura, Genta Kobayashi, Yoshio Kawamura.

**Methodology:** Yukio Nagano, Kei Kimura, Yoshio Kawamura.

**Project administration:** Yukio Nagano, Kei Kimura, Yoshio Kawamura.

**Resources:** Yoshio Kawamura.

**Software:** Yukio Nagano.

**Supervision:** Genta Kobayashi.

**Validation:** Yukio Nagano, Kei Kimura, Yoshio Kawamura.

**Visualization:** Yukio Nagano.

**Writing – original draft:** Yukio Nagano.

**Writing – review & editing:** Kei Kimura, Yoshio Kawamura.

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
