## [Decision Letter · Decision Letter 0]

15 Oct 2020

PONE-D-20-27527

Genomic diversity of 39 accessions of *Pyropia* species grown in Japan

PLOS ONE

Dear Dr. Nagano,

Thank you for submitting your manuscript to PLOS ONE. After careful consideration, we have decided that your manuscript does not meet our criteria for publication and must therefore be rejected.

I am sorry that we cannot be more positive on this occasion, but hope that you appreciate the reasons for this decision.

Yours sincerely,

Tzen-Yuh Chiang

Academic Editor

PLOS ONE

Reviewers' comments:

Reviewer's Responses to Questions

**Comments to the Author**

1. Is the manuscript technically sound, and do the data support the conclusions?

Reviewer #1: Partly

Reviewer #2: Partly

2. Has the statistical analysis been performed appropriately and rigorously? 

Reviewer #1: N/A

Reviewer #2: N/A

3. Have the authors made all data underlying the findings in their manuscript fully available?

Reviewer #1: No

Reviewer #2: Yes

4. Is the manuscript presented in an intelligible fashion and written in standard English?

Reviewer #1: Yes

Reviewer #2: Yes

5. Review Comments to the Author

Reviewer #1: This manuscript was already published in “bioRxiv – the preprint server for biology” under the title of “Genomic diversity of 39 Pyropia species grown in Japan (doi: https://doi.org/10.1101/2020.05.15.099044). This case is the redundant publication and I wonder the guideline of “PLOS ONE” can allow this case. I think that the nature of “bioRxiv” is not the process of the pre-review system and it is just one of the publication. Therefore, I cannot accept this manuscript for the publication

Reviewer #2: This manuscript analyzed the phylogenetic relations of 39 accessions of Pyropia species grown in Japan by chloroplast, mitochondrial and nuclear genome using high-throughput sequencing. In addition, the author also analyzed the genetic background of hybrid between P. yezoensis and P. tenera. The results of this study will be useful for breeding and the conservation of Pyropia species in Japan.

However, there are some problems existing in the structure and contents of the whole manuscript. The author attempted to elucidate the phylogenetic relationship of the samples from Japan. This part is the focus of the whole manuscript. In addition, the author also confirmed the hybrid relationship between between P. yezoensis and P. tenera, which is not consistent with the first part and can be deleted.

About the phylogenetic analysis, the author should make model test using different softwares, such as PAUP、PYLIP and Bayesian. Otherwise, the author referred that the topological structure of phylogenetic trees based on chloroplast, mitrochondrial and nuclear genome sequences are not scientific. Furthermore, the conclusion about the evoltionary history of chloroplast and mitrochondrial genome is not identical is also not scientific. So the author should analyze the evolutionary speed of chloroplast and mitrochondrial genome.

The author should show the detailed information about the sequences from chloroplast and mitrochondrial genome used in constructing phylogenetic tree in the method section.

The author should explain the reasons why China cultivar MK695880 clustered with Japan samples in Fig2.

The taxonomy of some samples were confused by the author previously. They were corrected based this manuscript. Yet this is the mistake by the author in the previous research and is not similar with the wild sample identification using phylogenetic analysis. So these related analysis should be re-writen.

In the result section, I am not sure whether the author removed the sequences from the contaminant from bacterial nuclear genome. Or the author should clearly state the genome sequences used for constructing the phylogenetic tree were indeed mapped to the reference genome.

6. PLOS authors have the option to publish the peer review history of their article (what does this mean?). If published, this will include your full peer review and any attached files.

Reviewer #1: No

Reviewer #2: No

- - - - -

---

## [Author Response · Author response to Decision Letter 0]

1 Dec 2020

We thank the editorial team for reversing a decision that was made against the editorial policy of PLOS ONE.

We are grateful for the suggestions made by the reviewers to improve our manuscript. We have revised the manuscript according to these suggestions. Furthermore, we have provided a point-by-point response to the reviewers’ comments as attached file.

We trust that the manuscript has been improved satisfactorily and hope that it is now acceptable for publication in PLOS ONE.

---

## [Decision Letter · Decision Letter 1]

25 Jan 2021

PONE-D-20-27527R1

Genomic diversity of 39 accessions of *Pyropia* species grown in Japan

PLOS ONE

Dear Dr. Nagano,

Thank you for submitting your manuscript to PLOS ONE. After careful consideration, we feel that it has merit but does not fully meet PLOS ONE’s publication criteria as it currently stands. Therefore, we invite you to submit a revised version of the manuscript that addresses the points raised during the review process.

The MS entitled Genomic diversity of 39 accessions of Pyropia species grown in Japan describes scientific advances in its findings. The research is up to date and suggests the further improvement of the species. Data is analyzed properly and findings from the study are fact-based. However, as all the reviewers have raised concern about the novelty of the work (as a preprint is available on a website), we suggest to address the issue (as suggested below) along with addressing all the comments. Based on the reviewers' comments we strongly recommend the MS for Major revision.

A rebuttal letter that responds to each point raised by the academic editor and reviewer(s). You should upload this letter as a separate file labeled 'Response to Reviewers'.A marked-up copy of your manuscript that highlights changes made to the original version. You should upload this as a separate file labeled 'Revised Manuscript with Track Changes'.An unmarked version of your revised paper without tracked changes. You should upload this as a separate file labeled 'Manuscript'

We look forward to receiving your revised manuscript.

Kind regards,

Arun Kumar Jugran and Berthold Heinze

Academic Editors

PLOS ONE

https://journals.plos.org/plosone/s/file?id=ba62

/PLOSOne_formatting_sample_title_authors_affiliations.pdf

2. In your Methods section, please provide additional details regarding the blades used in your study and ensure you have described the source. For more information regarding PLOS' policy on materials sharing and reporting, see https://journals.plos.org/plosone/s/materials-and-software-sharing#loc-sharing-materials.

3. We note that you are reporting an analysis of a microarray, next-generation sequencing, or deep sequencing data set. PLOS requires that authors comply with field-specific standards for preparation, recording, and deposition of data in repositories appropriate to their field. Please upload these data to a stable, public repository (such as ArrayExpress, Gene Expression Omnibus (GEO), DNA Data Bank of Japan (DDBJ), NCBI GenBank, NCBI Sequence Read Archive, or EMBL Nucleotide Sequence Database (ENA)). In your revised cover letter, please provide the relevant accession numbers that may be used to access these data. For a full list of recommended repositories, see http://journals.plos.org/plosone/s/data-availability#loc-omics or http://journals.plos.org/plosone/s/data-availability#loc-sequencing.

4. We note that Figure 1 in your submission contain map images which may be copyrighted. All PLOS content is published under the Creative Commons Attribution License (CC BY 4.0), which means that the manuscript, images, and Supporting Information files will be freely available online, and any third party is permitted to access, download, copy, distribute, and use these materials in any way, even commercially, with proper attribution. For these reasons, we cannot publish previously copyrighted maps or satellite images created using proprietary data, such as Google software (Google Maps, Street View, and Earth). For more information, see our copyright guidelines: http://journals.plos.org/plosone/s/licenses-and-copyright.

4.1. You may seek permission from the original copyright holder of Figure 1 to publish the content specifically under the CC BY 4.0 license.

4.2. If you are unable to obtain permission from the original copyright holder to publish these figures under the CC BY 4.0 license or if the copyright holder’s requirements are incompatible with the CC BY 4.0 license, please either i) remove the figure or ii) supply a replacement figure that complies with the CC BY 4.0 license. Please check copyright information on all replacement figures and update the figure caption with source information. If applicable, please specify in the figure caption text when a figure is similar but not identical to the original image and is therefore for illustrative purposes only.

Additional Editor Comments (if provided):

1 - The authors are right in saying that submitting a pre-print to bioRxiv is actually encouraged by PLOS ONE. The best way forward (or the prefered and suggested one) from there, however, is to upload a revised version to bioRxiv and use the "direct transfer to journal" possibility there. In this way, the various versions of the manuscript stay tightly connected to each other. bioRxiv points later readers to the journal article, if the material gets published. So a bioRxiv item is not an independent publication on its own, and that's why PLOS ONE encourages it.

2 - The style of the manuscript is not according to PLOS ONE criteria in that Results and Discussion come before Material and Methods here. So this must be corrected by the authors.

3 - I found the abstract not as informative about what was actually done in this study, as opposed to the last paragraph of the introduction, which gives a far better overview of the study. So I would like to encourage the authors to re-write their abstract accordingly, so that it gives a clear story of what was done in this study.

4 - It would help for the understanding of the sampling strategy to categorize/summarize the samples as, e.g. numbers of varieties, hybrids, related species, et cetera, and to also summarize the geographical sampling sites. Readers not familiar with the geography of Japan currently have bit of a hard time following the text, even when refering to the geographical map. This should be addressed and the text improved, in my opinion.

5 - Moreover, for the international and diverse readership of PLOS ONE, it would help I think if the authors briefly give a definition of these terms - varieties, hybrids, pure accessions et cetera - for non-seaweed specialists (as myself). This is important because of the implications for genomic data, i.e., whether samples are homozygous or heterozygous - what is the expectation in the various categories (varieties, hybrids, wild accessions, ...)? I found this a bit confusing when the 'allodiploid' nature of the hybrid is mentioned and discussed. My understanding after reading the text is that normally, these species are diploid; while the hybrid must then be allotretraploid, and by sampling an originally haploid cell line (which, in this case, must be allodiploid, correct?) which then undergoes diploidization (correct?), in this case we arrive at an allotetraploid? So when they write "allodiploid" here, they really mean an F1 hybrid cell lineage?

Please clarify all this; maybe my assumptions are wrong, or different terminology is used in the seaweed world; but a clarification would greatly improve the manuscript.

6 - further, I have the following minor editorial remarks:

L 109 - ... addition, they can be crossed WITH each other[32]

L 122 - Thus, mitochondrial sequences are determinant for clustering in the previous report[26] - ? I do not understand the meaning of this sentence.

L 147 - The mean coverage of Pyr_35 (P. dentata), Pyr_44 (P. tenuipedalis), and Pyr_45 (P. haitanensis) was 2.7, 8.8, and 4.2, respectively. - Use the X sign to indicate coverage, which is a convention that is useful I think: 2.7 X et cetera.

L 173 - The Pyr_27 is also a pure accession,... - define what "pure accession" means. See remark 5 just above.

L 194 - Of 34 accessions of P. yezoensis, 31 accessions were pure,so they should be homozygous. - define "pure" as opposed to (or identical to ?) "homozygous". Again, see remark 5 above.

L 300 - In contrast, there were two types of rRNA genes in the chloroplast genome. - Have the auhors considered the possibility of nuclear and/or mitochondrial copies of chloroplast rRNA genes? There are many publications on this phenomenon in the plant world. Or is this another peculiarity in the seaweed world that chloroplasts are "added" when species hybridize? Or is it just simply unknown (as they also point out that the mode of inheritance of organelles is not known in this group of seaweeds, at another location in the manuscript)?

L 367 - GetOrganelle - which reference genomes or 'baits' (if any?) were used to extract and map (assemble) the chloropalsts and mitochondria to?

"target genome(s) or sequence(s) as the seed" are used in GetOrganelle, when I look at the description of this programme.

Reviewers' comments:

Reviewer's Responses to Questions

**Comments to the Author**

1. If the authors have adequately addressed your comments raised in a previous round of review and you feel that this manuscript is now acceptable for publication, you may indicate that here to bypass the “Comments to the Author” section, enter your conflict of interest statement in the “Confidential to Editor” section, and submit your "Accept" recommendation.

Reviewer #3: All comments have been addressed

Reviewer #4: All comments have been addressed

2. Is the manuscript technically sound, and do the data support the conclusions?

Reviewer #3: Partly

Reviewer #4: Partly

3. Has the statistical analysis been performed appropriately and rigorously? 

Reviewer #3: Yes

Reviewer #4: I Don't Know

4. Have the authors made all data underlying the findings in their manuscript fully available?

Reviewer #3: Yes

Reviewer #4: Yes

5. Is the manuscript presented in an intelligible fashion and written in standard English?

Reviewer #3: No

Reviewer #4: Yes

6. Review Comments to the Author

Reviewer #3: This research paper is already available in public database/repository, so in the current scenario this unethical to publish. Paper doesn’t have authenticity and novelty. It may resubmit to the journal only after withdrawal from bioRxiv preprint server (Cold spring harbor laboratory) https://www.biorxiv.org/content/10.1101/2020.05.15.099044v1.full.

Nevertheless, the manuscript submitted to the Journal entitled “Genomic diversity of 39 accessions of Pyropia species grown in Japan" is an informative piece of work and seems useful in view of harnessing the potential of these important marine algae in breeding and genetics. Although, authors have included limited (39) number of accessions in the study, work may help to breeders and researcher to generate the information for identification of best planting material.

Based on my observations I recommend the manuscripts for major revision. To improve the manuscript quality, the following comments and suggestions are required to be addressed;

The economic importance and medicinal values of the Pyropia species is lacking in the introduction section of the manuscript.

Why the assessment of the genetic diversity is important within the the Pyropia species. Explain in introduction in brief.

What is Ariake sound, mention clearly in M&M of MS.

In material and methods part of the MS, the construction of DNA library is missing, include some lines about that.

How organellar (mitochondrial and chloroplast) DNA was extracted..?

What type of data was used to construct a phylogenetic tree, need to clearly mention in materials and methods.

Discuss the results that mitochondrial DNA sequences couldn’t revealed genetic variability among the of 39 accessions of Pyropia species; What may be the probable cause of that.

And also explain how about the chloroplast based genetic variations showed in gbPyropia species.

Nuclear DNA based genetic analysis was significant in compared to that of organellar genetic diversity, what may be the possible reasons for this distinctive nature of between organellar and nuclear genetics.

Line 403 is the repeat of 402; these can be clubbed in a single sentence.

In results section, lines 73 -75 should be part of materials methods.

There are discussions parts mingled with results which enlarged the size of results, check thoroughly and trim the results section.

Reviewer #4: Dear authors,

The research work presented in the manuscript detailed about genetic diversity of 39 accessions of Pyropia species grown in Japan.The study seems important. In my view the work will be of more significance if the authors could make out correlation of phenotypic differences with their genotypes. The results are organized into headings which does not go well, the results of sequencing are not mentioned in the results section. In the M&M section the authors wrote that DNA and RNA was isolated but in the heading they have mentioned DNA sequencing only. No clarity about what was sequenced. Details need to be incorporated.

7. PLOS authors have the option to publish the peer review history of their article (what does this mean?). If published, this will include your full peer review and any attached files.

Reviewer #3: **Yes: **Ram Baran Singh

Reviewer #4: No

---

## [Author Response · Author response to Decision Letter 1]

7 Mar 2021

Please check "Response to reviewers", because we have many responses.

---

## [Decision Letter · Decision Letter 2]

4 May 2021

PONE-D-20-27527R2

Genomic diversity of 39 samples of *Pyropia* species grown in Japan

PLOS ONE

Dear Dr. Nagano,

Thank you for submitting your manuscript to PLOS ONE. After careful consideration, we feel that it has merit but does not fully meet PLOS ONE’s publication criteria as it currently stands. Therefore, we invite you to submit a revised version of the manuscript that addresses the points raised during the review process.

Address the minor edit suggested by Reviewer #4 to "merge the DNA purification and DNA sequencing sub-sections" and to rewrite/reword the DNA purification step as described by the reviewer (i.e., the procedure is for purifying DNA - no need to indicate RNA other than the procedure includes RNAase to get rid of the RNA).

We look forward to receiving your revised manuscript.

Kind regards,

Randall P. Niedz

Academic Editor

PLOS ONE

Journal Requirements:

Reviewers' comments:

Reviewer's Responses to Questions

**Comments to the Author**

1. If the authors have adequately addressed your comments raised in a previous round of review and you feel that this manuscript is now acceptable for publication, you may indicate that here to bypass the “Comments to the Author” section, enter your conflict of interest statement in the “Confidential to Editor” section, and submit your "Accept" recommendation.

Reviewer #4: All comments have been addressed

Reviewer #5: All comments have been addressed

2. Is the manuscript technically sound, and do the data support the conclusions?

Reviewer #4: Partly

Reviewer #5: Yes

3. Has the statistical analysis been performed appropriately and rigorously? 

Reviewer #4: I Don't Know

Reviewer #5: Yes

4. Have the authors made all data underlying the findings in their manuscript fully available?

Reviewer #4: Yes

Reviewer #5: Yes

5. Is the manuscript presented in an intelligible fashion and written in standard English?

Reviewer #4: Yes

Reviewer #5: Yes

6. Review Comments to the Author

Reviewer #4: Authors should merge the DNA purification and DNA sequencing sub-sections and I recommend to rewrite the DNA purification step in a proper way. When DNA is isolated RNAase treatment is done to get rid of RNA contamination, but that need not be written as DNA and RNA is isolated, when your aim is to isolate or purify only DNA.

Reviewer #5: Review of the manuscript PONE-D-20-27527 entitled: “Genomic diversity of 39 samples of Pyropia species grown in Japan”

The submitted manuscript is an article that presents a phylogenetic analysis of 39 samples of Pyropia yezoensis, an important marine crop that is used as ingredients of sushi and snacks around the world. The Authors using HTS methods were able to perform a deep study of molecular variability of P. yezoensis isolated from different localities.

In my opinion, the revised paper after several corrections is well written and introduces the reader to the subject matter presented here. The methodology has been selected and presented correctly. The description of the results is presented clearly. The discussion section is revised according to reviewers' comments.

The main complaint relating to an earlier publication of the text in the bioRxiv repository has already been explained.

Therefore, I believe that it is a very valuable text which in revised form can be published in PLOS ONE.

7. PLOS authors have the option to publish the peer review history of their article (what does this mean?). If published, this will include your full peer review and any attached files.

Reviewer #4: No

Reviewer #5: No

---

## [Author Response · Author response to Decision Letter 2]

6 May 2021

Reviewer #4: Authors should merge the DNA purification and DNA sequencing sub-sections and I recommend to rewrite the DNA purification step in a proper way. When DNA is isolated RNAase treatment is done to get rid of RNA contamination, but that need not be written as DNA and RNA is isolated, when your aim is to isolate or purify only DNA.

We appreciated your helpful comments.

We have merged the two sections and changed the title to "DNA purification and sequencing".

In addition, we have changed the sentence about DNA purification as follows: “DNA was extracted from the conchocelis of each sample using the DNAs-ici!-F (Rizo, Tsukuba, Japan) according to the instructions of the manufacturer, followed by RNase A (NIPPON GENE, Tokyo, Japan) treatment.”

Reviewer #5: Review of the manuscript PONE-D-20-27527 entitled: “Genomic diversity of 39 samples of Pyropia species grown in Japan”

The submitted manuscript is an article that presents a phylogenetic analysis of 39 samples of Pyropia yezoensis, an important marine crop that is used as ingredients of sushi and snacks around the world. The Authors using HTS methods were able to perform a deep study of molecular variability of P. yezoensis isolated from different localities.

In my opinion, the revised paper after several corrections is well written and introduces the reader to the subject matter presented here. The methodology has been selected and presented correctly. The description of the results is presented clearly. The discussion section is revised according to reviewers' comments.

The main complaint relating to an earlier publication of the text in the bioRxiv repository has already been explained.

We are encouraged by your comments as we have made repeated revisions (*^-^*)

Additional Changes

In addition to the changes mentioned above, we have made the following changes. We apologize for these corrections.

1. We made some minor changes, such as spelling errors.

2. We have moved the legend in Figure 3 to the appropriate position.

3. The internal title of the PDF file of S9 Fig was S2 Fig, so we changed it to S9 Fig.

4. There was a mistake in Figure 8 that we failed to fix in the previous revision. In order to fit the figure to the manuscript, we changed the strain name from "S-18" to "Shin Saga 4 gou" in Figure 8.

---

## [Editor Report · Decision Letter 3]

12 May 2021

Genomic diversity of 39 samples of *Pyropia* species grown in Japan

PONE-D-20-27527R3

Dear Dr. Nagano,

We’re pleased to inform you that your manuscript has been judged scientifically suitable for publication and will be formally accepted for publication once it meets all outstanding technical requirements.

Kind regards,

Randall P. Niedz

Academic Editor

PLOS ONE
---

## [Editor Report · Acceptance letter]

17 May 2021

PONE-D-20-27527R3 

Genomic diversity of 39 samples of *Pyropia* species grown in Japan 

Dear Dr. Nagano:

I'm pleased to inform you that your manuscript has been deemed suitable for publication in PLOS ONE. Congratulations! Your manuscript is now with our production department. 

Kind regards, 

on behalf of

Dr. Randall P. Niedz 

Academic Editor

PLOS ONE